# A Novel Technique for Disinfection Treatment of Contaminated Dental Implant Surface Using 0.1% Riboflavin and 445 nm Diode Laser—An In Vitro Study

**DOI:** 10.3390/bioengineering9070308

**Published:** 2022-07-12

**Authors:** Luka Morelato, Ana Budimir, Igor Smojver, Ivan Katalinić, Marko Vuletić, Muhamed Ajanović, Dragana Gabrić

**Affiliations:** 1Faculty of Dental Medicine, University of Rijeka, 51000 Rijeka, Croatia; luka.morelato@fdmri.uniri.hr; 2Department of Clinical and Molecular Microbiology, School of Medicine, University Hospital Centre Zagreb, University of Zagreb, 10000 Zagreb, Croatia; abudimir@kbc-zagreb.hr; 3Private Hospital St. Catherine, 10000 Zagreb, Croatia; ismojver@gmail.com (I.S.); ivan.katalinic@svkatarina.hr (I.K.); 4Department of Oral Surgery, School of Dental Medicine, University Hospital Centre Zagreb, University of Zagreb, 10000 Zagreb, Croatia; mvuletic@sfzg.hr; 5Clinic for Dental Prosthetics, Faculty of Dentistry with Clinics, University of Sarajevo, 71000 Sarajevo, Bosnia and Herzegovina; ajanovic@bih.net.ba

**Keywords:** decontamination, dental implants, diode laser, photodynamic therapy, riboflavin

## Abstract

Background: Antimicrobial photodynamic therapy (PDT) has been introduced as a potential option for peri-implantitis treatment. The aim of this study is to evaluate the antimicrobial effect of a novel technique involving a combination of 445 nm diode laser light with 0.1% riboflavin solution (used as a photosensitizing dye) as applied on a bacterial–fungal biofilm formed on implants and to compare the performance of this technique with that of the commonly used combination of 660 nm diode laser with 0.1% methylene blue dye. Methods: An in vitro study was conducted on 80 titanium dental implants contaminated with *Staphylococcus aureus* (SA) and *Candida albicans* (CA) species. The implants were randomly divided into four groups: negative control (NC), without surface treatment; positive control (PC), treated with a 0.2% chlorhexidine (CHX)-based solution; PDT1, 660 nm (EasyTip 320 µm, 200 mW, Q power = 100 mW, 124.34 W/cm^2^, 1240 J/cm^2^) with a 0.1% methylene blue dye; and PDT2, 445 nm (EasyTip 320 µm, 200 mW, Q power = 100 mW, 100 Hz, 124.34 W/cm^2^, 1.24 J/cm^2^) with a 0.1% riboflavin dye. Results: The PDT1 and PDT2 groups showed greater reduction of SA and CA in comparison to the NC group and no significant differences in comparison to the PC group. No statistically significant differences between the PDT1 and PDT2 groups were observed. Conclusions: A novel antimicrobial treatment involving a combination of 445 nm diode laser light with riboflavin solution showed efficiency in reducing SA and CA biofilm formation on dental implant surfaces comparable to those of the more commonly used PDT treatment consisting of 660 nm diode laser light with methylene blue dye or 0.2% CHX treatment.

## 1. Introduction

With the discovery of osseointegration and the associated technological development, dental implant therapy is being increasingly used to replace missing teeth. With the increase in the number of dental implants, the frequency of complications in clinical practice, in the form of peri-implant diseases, has also increased [1].

More than 20 different types of micro-organisms have been identified as plaque components occurring around implants, which lead to peri-implant soft tissue inflammation, where the progression of inflammation can cause peri-implant bone resorption [2,3]. The bacterial composition in subgingival peri-implant pockets around the implant was found to be highly correlated with characteristics of the periodontal pocket microflora [4]. *Staphylococcus aureus*, although not a typical periodontal pathogen, has been listed as a key microorganism that causes peri-implantitis [5,6]. As *S. aureus* has a high affinity for the titanium surface of implants, it has been recognized as an early colonizer and is strongly associated with biofilm formation on titanium implants [7]. Furthermore, the presence of high levels of *S. aureus* in deep peri-implant pockets has also been associated with the presence of suppuration and bleeding upon probing [8].

Unlike many bacterial species indicated to cause peri-implantitis, *Candida* species, which are the most commonly isolated fungi in peri-implant diseases, develop thick biofilms over the implant surface [9]. The effects of *Candida* species fungi have already been associated with the maintenance of periodontal inflammation in periodontitis [10]. If peri-implantitis is not diagnosed in time and adequately treated, implant loss may occur [11].

The treatment of peri-implantitis is a great challenge for clinicians due to the complex pathogenesis of the disease and the difficulty in decontaminating the rough surface of dental implants [12,13,14]. Although various therapies have been proposed in the literature, non-surgical manual debridement is still considered the gold standard for the treatment of peri-implant inflammation [15]. In addition to mechanical debridement methods, chemical techniques for the decontamination of the implant surface are used, including rinsing surfaces with chlorhexidine, antibiotics, and hydrogen peroxide [14]. However, none of the well-known methods can completely remove or inactivate peri-implant pathogens due to various anatomical relationships and the complexity of the implant surface [16]. Furthermore, it is possible to damage the implant surface microstructure using certain agents [17]. Adjunctive therapies have been extensively investigated in contemporary dentistry, and many in vitro, animal, and clinical studies have already shown that non-invasive photodynamic therapy (PDT) can serve as a successful and safe adjunctive therapeutic protocol combined with prior mechanical cleaning of the implant thread surface for the treatment of peri-implantitis [18,19,20].

PDT implant surface treatment is based on a photochemical reaction consisting of photosensitizers, oxygen, and light. After binding photosensitive molecules to target cells, the implant surface is irradiated with light at a certain wavelength in the presence of oxygen. The excited photosensitizers can undergo type I (electron transfer) and/or type II (energy transfer) reactions to produce reactive oxygen species (ROS), resulting in disruption of the bacterial cell wall and/or normal metabolism, leading to bacterial cell damage or death [21]. The described mechanisms do not cause host cell damage, as the human cell has mechanisms to survive oxidative stress, such as those involving catalase and superoxide dismutase enzymes [22]. The development of microbial resistance to PDT is not probable, as the bactericidal effect is achieved through the action of oxygen radicals on the cellular components of the micro-organisms [23].

Various photosensitizers have been used in clinical practice, such as toluidine blue, methylene blue, curcumin, and riboflavin, which can be activated by light at appropriate wavelengths. Studies have shown that in vitro photosensitization of bacteria can be achieved without any damage to the treated dental implant surfaces [24]. Although numerous studies have been conducted on the decontamination of dental implant surfaces using PDT, there is still no consensus in the literature regarding which particular PDT radiation parameters (wavelength) or photosensitizers are the most effective at reducing the presence of bacteria on dental implant surfaces. In clinical practice and studies, most research employing photodynamic therapy has used methylene or toluidine blue in combination with 600–660 nm wavelength light from the red part of the spectrum. Despite often use of methylene or toluidine blue as a photosensitizer, it has disadvantages such as discoloration of surrounding soft tissue. In a systematic review conducted by the American Academy of Periodontology which evaluated the efficacy of PDT with methylene or toluidine blue concluded that PDT may provide similar clinical improvements in PD and CAL when compared with conventional periodontal therapy for both periodontitis and peri-implantitis patients [25]. However, a novel method that combines 0.1% riboflavin as a photosensitizer and a laser beam with blue light (in particular, of 445 nm wavelength) was recently introduced. This protocol was tested on *S. Aureus*, *C. Albicans* and *Enterococcus faecalis* in endodontics for the first time by Katalinic et al., with promising results [26].

Therefore, the aim of this study was to evaluate the antimicrobial effect of the novel combined therapy using 445 nm diode laser light and 0.1% riboflavin solution (as photosensitizing dye) on implant biofilm, and its performance was compared to that of 660 nm diode laser light with methylene blue dye, which is more commonly used in PDT.

## 2. Material and Methods

The study was conducted according to the guidelines of the Declaration of Helsinki and approved by the Institutional Review Board (or Ethics Committee) of the School of Dental Medicine, University of Zagreb (05-PA-30-XII-12/2019 on 5 December 2019).

The in vitro study was conducted on 80 titanium dental implants (GC Aadva Standard Implants; GCTech.Europe GmbH, Breckerfeld, Germany) of 4.0 mm in diameter and 10 mm in length. All microbiological procedures were performed at the laboratory of the Department of Clinical and Molecular Microbiology, University Hospital Centre Zagreb.

For the determination of significant differences in numerical values between measurements at the level of 0.05 and the power set to 0.8 (i.e., a large effect size of 0.8), the minimum required sample size was calculated as 20 subjects per group.

*Staphylococcus aureus* and *Candida albicans* strains, isolated from clinical samples at the University Hospital Center Zagreb, were used for contamination of the dental implants. The oral cavity swabs taken from the patient with peri-implantitis were inoculated on a blood agar plate (Columbia agar plate with 5% of sheep blood), colonies with distinct morphology were identified as *S. aureus* by use of matrix-assisted laser desorption/ionization time-of-flight (MALDI TOF) technology, and the susceptibility testing was performed according to EUCAST standards. For mycology samples, swabs were inoculated to Saburaud plate and glucose broth, and after the incubation at a temperature of 28°C for 48 h, colonies were identified by MALDI TOF as *Candida albicans*. The bacteria and fungi were grown separately in Columbia Agar for 72 h. Then, a separate brain heart infusion broth suspension of each of the micro-organisms was prepared, and these were then mixed to form a joint suspension. An optical densitometer (Densimat, Biomerieux, Marcy l’Etoile, France) was used to set a density of 600 nm, which is equivalent to a 1 × 10^8^ colony forming units per milliliter (CFU/mL). The suspension concentration was adjusted to 0.5 according to the McFarland standard. All dental implants were removed from sterile packaging and then immersed for 14 days, under aerobic conditions, in 300 µL of the mixed bacterial–fungal suspension (containing *S. aureus* and *C. albicans* at a density of 0.5 McFarland units). Suspension of microorganisms was prepared and added every 48–72 h.

Implants were removed from the bacterial–fungal suspension using sterile forceps, gently dried with sterile gauze to remove the excess bacterial and fungal solution, and placed on sterile silicone holders (Zetaplus, Zhermack, S.p.A., Badia Polesine, Italy) to prevent rotation and movement during decontamination and sample taking (Figure 1). The formation of bacterial and fungal plaques was confirmed through SEM (JSM-7800 F Schottky Field Emission Scanning Electron Microscope, JEOL Ltd., Tokyo, Japan); see Figure 2.

The implants were randomly divided into four groups (*n* = 20) based on the planned surface treatment after contamination and biofilm formation, as follows:

The negative control group (NC) did not undergo surface treatment. In the positive control group (PC), after placing the implant in the sterile holder, the implant surface was treated with a sterile cotton pellet soaked in a 0.2% chlorhexidine-based solution (Curasept ADS 220, Curaden AG, Kriens, Switzerland) for 60 s using a brushing movement in the direction of the implant threads (Figure 3). For the contaminated implants of the first photodynamic therapy group (PDT1), the surface was prepared using a similar procedure, whereby 0.1% methylene blue dye solution (as a photosensitizer) was prepared in a hospital pharmacy that is used in both general medicine and dentistry and was applied using a sterile syringe for 60 s followed by gentle washing with a sterile saline solution. The surface was then dried with sterile gauze and treated with laser (SiroLaser Blue, Dentsply Sirona, Bensheim, Germany) light at 660 nm (Q power = 100 mW), in continuous-wave mode using an EasyTip 320 µm, with the max power density of 124.34 W/cm^2^ and max energy density of 1240 J/cm^2^, for 60 s moving in circles at approximately 1 mm from the implant surface. This process was carried out by one skilled operator. The surface of the contaminated implants of the second photodynamic therapy group (PDT2) was treated with 0.1% riboflavin dye (same concentration of methylene blue dye solution in PDT1 group) prepared in a hospital pharmacy that is used in general medicine and applied using a sterile syringe and left for 60 s, followed by washing with sterile saline solution and gentle drying with sterile gauze (Figure 4). The treated implant surface was then radiated with a diode laser (SiroLaser Blue, Dentsply Sirona, Bensheim, Germany) at 445 nm (Q power = 100 mW) in the pulsed mode of 100 Hz using an EasyTip 320 µm, with the same max power density of 124.34 W/cm^2^ as the compared 660 nm laser but different energy density of 1.24 J/cm^2^, for 60 s moving in circles at approximately 1 mm from the implant surface. The same skilled operator also carried out this process (Figure 5).

Samples were collected using sterile plastic inoculating loops, with five horizontal brushing movements between the second and fifth implant threads. The samples were then immersed in 0.5 mL of sterile brain heart infusion. Each tube, containing the infusion and plastic inoculating loop, was then vortexed (Corning LSE vortex mixer, Corning, NY, USA) for 60 s (Figure 6).

Serial dilutions were performed in 96 microtiter microplates; 20 µL of suspension was added to 180 µL of nutrient broth and mixed, and 20 µL was transferred to the next well. Serial dilution up to 8 dilutions were performed. From each well, we transferred 50 µL to a blood agar plate. After an incubation period of 48 h at 37 °C, the CFUs were counted (Figure 7).

Macroscopically distinct colonies were confirmed using an MALDI Biotyper (Bruker Daltonics, Hamburg, Germany), and the obtained results were entered into the prepared tables.

One randomly chosen sample from each group was subjected to SEM to assess the reduction in bacterial and fungal cell growth (Figure 8, Figure 9, Figure 10 and Figure 11).

### Statistical Analysis

Numerical data were described in terms of median and interquartile range. The normality of the distribution of continuous variables was assessed using the Shapiro–Wilk test. The Mann–Whitney U test (with Hodges–Lehmann median difference) was used to compare the medians between pairs of groups. All p-values were two-sided. The level of significance was set at alpha = 0.05. The statistical analysis was performed using MedCalc^®^ Statistical Software version 20.023 (MedCalc Software Ltd., Ostend, Belgium; https://www.medcalc.org; accessed on 11 November 2021).

## 3. Results

Regarding the tested micro-organism species, the results obtained for control groups (NC and PC) were compared with those from the treatment groups (PDT1 and PDT2) and considered individually for both *S. aureus* (Table 1, Figure 12) and *C. albicans* (Table 2, Figure 13). A comparison between the treatment groups PDT1 and PDT2 for both *S. aureus* and *C. albicans* is shown in Table 3.

Significant differences were found when comparing the negative control group (NC), with no treatment used, to the positive control group (PC) when using 0.2% CHX treatment, in terms of the reduction in dental implant surface contamination by *S. aureus* and *C. albicans*, according to the micro-organism CFU counts as shown in Table 1 and Table 2. Results in Table 1 and Table 2 show that the first photodynamic therapy group (PDT1), in which the photosensitizer was 0.1% methylene blue activated by a 660 nm diode laser, and the second photodynamic therapy group (PDT2), with 0.1% riboflavin dye and a 445 nm diode laser, both showed greater reductions in SA and CA CFU counts when compared with those of the NC group, but no significant difference when they were compared with those of the PC group. Results from Table 1 and Table 2 were summarized and compared in Table 3, and no statistically significant differences between the PDT1 and PDT2 groups in terms of the CFU counts of *S. aureus* and *C. albicans* after an incubation period of 48 h were observed.

Dense, evenly distributed microorganism biofilm formation is visible in SEM analysis of the NC group. In the PC, PDT1, and PDT2 groups, a few single or smaller groups of microorganisms can be found unevenly distributed on the rough implant surface. The vitality of the cell cannot be confirmed using SEM analysis.

## 4. Discussion

Decontamination of the implant surface is the main goal of peri-implantitis treatment [19]. Structural irregularities of differently treated surfaces on titanium dental implants, which play a role in promoting osseointegration, may also facilitate the bacterial accumulation and prevent complete decontamination [27]. Gustumhaugen et al., when testing the accumulation and removal efficiency of bacteria on titanium disks of different roughness, have shown that a smoother surface leads to lower levels of biomass retention after chemical and mechanical debridement [28]. In addition, designing dental implants in the form of threads of different expressions and depths makes it more difficult to access and thoroughly clean all implant surfaces [27]. According to previous statements, tests conducted on real dental implants, although being more expensive, should yield more precise results than studies conducted on titanium discs, as the surface micro-and macromorphology will be the same as that in clinical situations. When carrying out photodynamic therapy, the use of photosensitive agents for the treatment of periodontitis or peri-implantitis has various advantages. These materials can penetrate the pores of the implant and root surfaces that are not accessible when using a mechanical protocol [24]. According to Deppe et al., photodynamic therapy has no adverse effects on the implant surface, but thermal changes may vary between different types of implant surface coatings. The same author concluded that using a 445nm diode laser in continuous wave (CW) mode (≥0.8 W), and higher power outputs (≥1.0 W) in pulsed mode for longer exposure times (>10 s) may cause harmful rises in temperature of more than 10 °C. However, both CW and pulsed laser irradiation can prevent harmful rises in temperature when used at moderate parameters (1 W, 10 s, 50% DC or 3 W, 20 s, 10% DC). [29]. Knowing this, the 445 nm laser (320 µm, 60 s, 200 mW, Q power = 100 mW, 100 Hz, 124.34 W/cm^2^, 1.24 J/cm^2^) used in the study with an output power of 100 mW is safe to use, and pulsed mode makes its use more secure and prevents potential rises in temperature.

Nevertheless, the constant moving of the fiber inside the peri-implant pocket is also important to keeping the temperature rise as low as possible to avoid potential thermal damage to the neighboring tissues [30]. 

In the present study, 0.2% chlorhexidine (CHX) was used as a positive control, which was applied using cotton pellets using a brushing movement, thus simulating in vivo usage. No statistically significant differences were demonstrated when compared to PDT. Widodo et al. reported that compared to PDT with methylene blue dye and a 660 nm laser, treatment with 0.2% CHX had a lower effect on implant surfaces of different roughness contaminated with *S. aureus* [31]. The difference in the obtained results is likely because Widodo et al., in their study, immersed titanium discs into chlorhexidine solution without cotton pellet mechanical surface treatment. This conclusion is also supported by their subsequent results in which they reported a more effective bacterial load reduction using brushing movements with a sterile cotton pellet alone (without CHX) on a smooth implant surface. However, the antimicrobial activity results for CHX should be taken with caution given its properties of binding to the implant surface with a subsequent gradual release, thus producing a bactericidal effect lasting up to 24 h after rinsing, in addition to the possibility of introducing CHX into the medium, resulting in a sustained bactericidal effect [32].

Various bacteria show different sensitivities to light at certain wavelengths due to direct effects on the bacteria, which can produce endogenous photosensitizers, and the affinity of bacterial absorption toward different photosensitizers [33]. Bärenfaller et al. showed that blue light had the same effect on the periodontal pathogens *Porphyromonas gingivalis* and *Prevotella intermedia* when used in the absence of photosensitizers and concluded that the endogenous substances μ-oxo bis and hematin produced by the bacteria themselves acted as photosensitizers which killed the bacteria in their experiments [34]. Unlike blue light, the use of red light did not result in strong reductions in these periodontal pathogens. Previous studies have reported moderate effects of red light on *P. intermedia* and *P. gingivalis*, as the absorption of endogenous porphyrins in the red spectrum is relatively low [35]. All before mentioned periodontal pathogens are anaerobic bacterial species, which require special equipment to obtain anaerobic conditions for properly conducting the research, so further in vitro research has to go in that direction.

Knowing this, in our study, we chose to use transparent *Staphylococcus* bacteria species to exclude the potential effects of the light itself (red or blue). Similar to the previously mentioned black-pigmented bacteria, elimination of *C. albicans* using blue light was shown to be highly efficacious, likely due to the presence of intracellular porphyrins and flavins [36,37]. To the best of our knowledge, there is no reported effect when using just red light on *C. albicans* species. Wiench et al. reported that due to the size of *Candida* cells and the presence of the cell wall, photosensitizer incubation time has an impact on PDT efficacy and suggested 7-10 min as the most efficient period [38]. Despite those findings, the application and incubation period of photosensitizing solution in the oral cavity with the salivary flow and long clinician chairside time makes this approach impractical in clinical practice.

In this study, riboflavin, as an innovative photosensitizing agent activated by diode laser blue light at 445 nm, was compared with methylene blue, as a photosensitizing agent activated by diode laser red light at 660 nm. Riboflavin (vitamin B2) is a vitamin naturally present in food, characterized by its yellow color and that it does not cause serious discoloration of teeth or surrounding tissue, unlike toluidine or methylene blue, and can therefore be used in the aesthetic zone [33]. The increase in ROS levels that results from riboflavin irradiation, using a blue light spectrum with a maximum absorption wavelength of ~450 nm, was shown to be 200% higher when compared with other yellow-colored agents activated by the same blue light source [39]. At the same time, ROS levels of riboflavin are more than five times lower than those of toluidine blue (0.51 vs. 2.70 ROS/µM) [40]. Although a study by Bärenfaller et al. has stated that the antimicrobial effect of riboflavin in combination with blue light is inferior to the effect of toluidine blue activated by red light [32], their results cannot be compared with the results of our study due to the use of different bacterial species, the study being performed with organisms in planktonic form rather than on implant surfaces, and the light source being an LED curing lamp which gives lower max power density per cm^2^ than laser light.

The concentration of riboflavin used in the study was chosen based on a concentration used in previous studies by Bärenfaller et al. Katalinic et al. The effect of different concentrations that can be more adequate could be tested in further studies. In the current study, the authors observed that riboflavin dye has poor solubility; it must be well shaken before usage. Implementing E101a (riboflavin-5′-phosphate) which is easier dissolved could be the answer to this problem. [26]

Leelanarathiwat et al. have tested flavin mononucleotide (FMN), which is an important cofactor of riboflavin with a similar chemical formula, in which the primary hydroxy group has been converted to its dihydrogen phosphate ester. FMN was tested as a PS for the decontamination of *S. aureus* on titanium sandblasted and acid-etched discs, activated by a blue high-power LED light (FotoSan^®^ BLUE LAD, CMS Dental APS, Copenhagen, Denmark) with wavelengths in the range of 450–470 nm. These were compared with MB activated using red light, and no differences were observed between the two PDT systems [41]. Although the light source was not a diode laser with a much higher power density (LED light 3.7–4 W/cm^2^ vs diode laser 124.34 W/cm^2^), as in our study, these results were in agreement with our study results from experiments conducted on dental implant surfaces contaminated with *S. aureus*. A study using the same bacteria and fungi species, and with a similar riboflavin–445 nm diode laser disinfection protocol, was conducted by Katalinic et al.; their results were similar to those of our study, but these results should be compared with caution, due to differences in the studied surface, i.e., of extracted tooth root canal vs. titanium dental implant [26].

In contrast to the use of riboflavin, for which there are only a few studies, there are numerous studies using MB and activation by red light, and these have shown positive antimicrobial results for in vivo and in vitro peri-implantitis treatment. Aside from the abovementioned blue coloring of the surrounding tissue, MB also shows a negative cytotoxic effect on fibroblasts (whether inactivated or activated by a light source); nonetheless, its cytotoxic effect is much lower than that of 2% CHX [42]. The antimicrobial effect of methylene blue, as a photosensitizer, on *C. albicans* has been demonstrated in various studies [43,44]. In the current study, we found that the effect of 445 nm light in combination with riboflavin was similar to the decontamination effect of methylene blue in combination with 660 nm light. Several studies have confirmed the antibacterial activity of MB against *S. aureus* biofilms using red laser photoactivation [45,46]. Studies have also shown that prolonged application of PDT in combination with methylene blue, with treatment from 1 to 5 min, can lead to a significant reduction in biofilm viability [47]. During clinical work, the possibility of prolonged PDT use, due to the time spent by medical staff, is questionable; therefore, no extended protocols (5 min) were tested in this study. It should be noted that the achieved effects were obtained after a single 60 s treatment, which suggests that repeated treatment could yield more positive effects.

In a review by Alasqah et al., all of the included in vitro PDT studies, despite using different bacteria, implant surfaces, photosensitizers, and wavelengths, showed significant reductions in the number of bacteria. In the same review, the authors also concluded that PDT cannot disrupt biofilm on dental implant surfaces but can reduce bacterial viability, as additional SEM analysis showed the presence of bacteria on the titanium surface, regardless of the disinfection treatment [48]. In the present study, as in the other studies, SEM indicated the presence of bacteria on the titanium surface, regardless of the disinfection treatment. Therefore, we can support the conclusion that PDT could play a vital role in reducing the bacterial load around dental implants as an additional surface disinfection technique to complement mechanical debridement [49].

Finally, the results of the presented in vitro studies may not be fully transferred or applied to in vivo conditions. Environmental factors, such as limited accessibility, plaque accumulation, salivation, immune system action, and so on, cannot be established using in vitro studies. Within the limitations of this in vitro study, these findings encourage further investigations into the in vitro and in vivo effects of novel disinfection therapies using riboflavin and 445 nm blue light.

## 5. Conclusions

Despite the limitations of this study, we conclude that the novel PDT using riboflavin and blue light at 445 nm led to a statistically significant reduction in viable micro-organisms, comparable to that achieved by 0.2% CHX disinfection. Following the presented results, the tested laser protocol could be recommended for clinical use in peri-implantitis therapy, especially in the aesthetic zone, as the light-yellow color of riboflavin does not interfere with aesthetics, which can be the case for methylene blue. However, we can only recommend the novel therapy as an adjunct to the mechanical cleaning of the implant surface, as it cannot completely eradicate all micro-organisms and organic material.

## Figures and Tables

**Figure 1 bioengineering-09-00308-f001:**
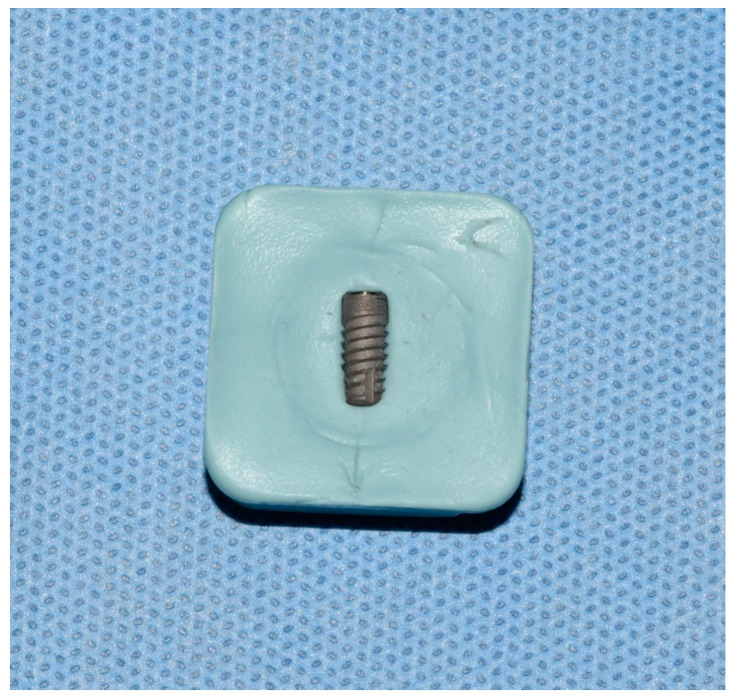
Contaminated dental implant in sterile silicon holder.

**Figure 2 bioengineering-09-00308-f002:**
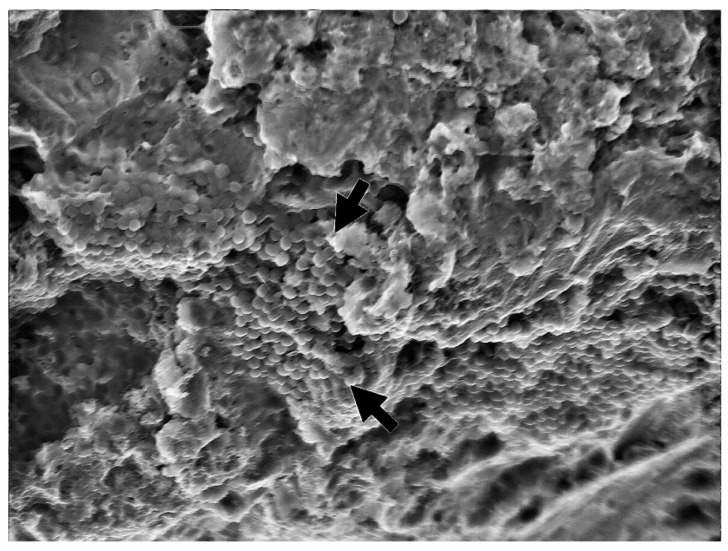
Biofilm formation confirmed using SEM (3000×)—arrows showing microorganism biofilm formation.

**Figure 3 bioengineering-09-00308-f003:**
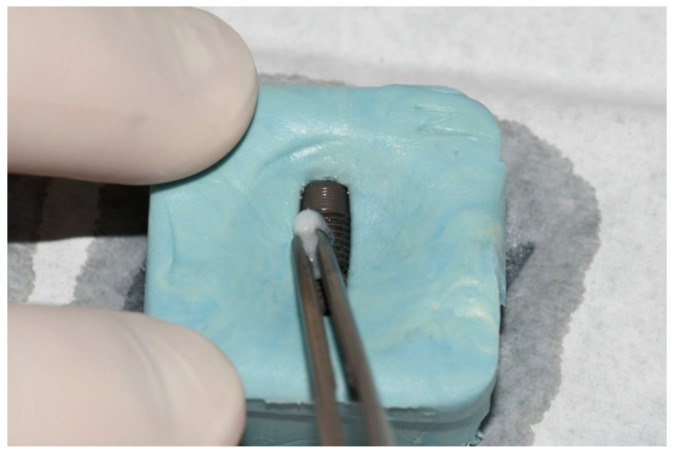
Decontamination of implant surface with sterile cotton pellet soaked in 0.2% chlorhexidine-based solution.

**Figure 4 bioengineering-09-00308-f004:**
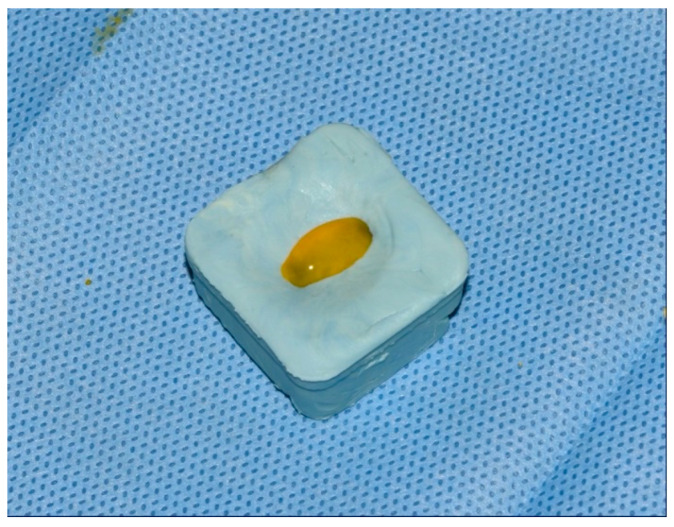
Treatment of implant with 0.1% riboflavin dye for 60 s.

**Figure 5 bioengineering-09-00308-f005:**
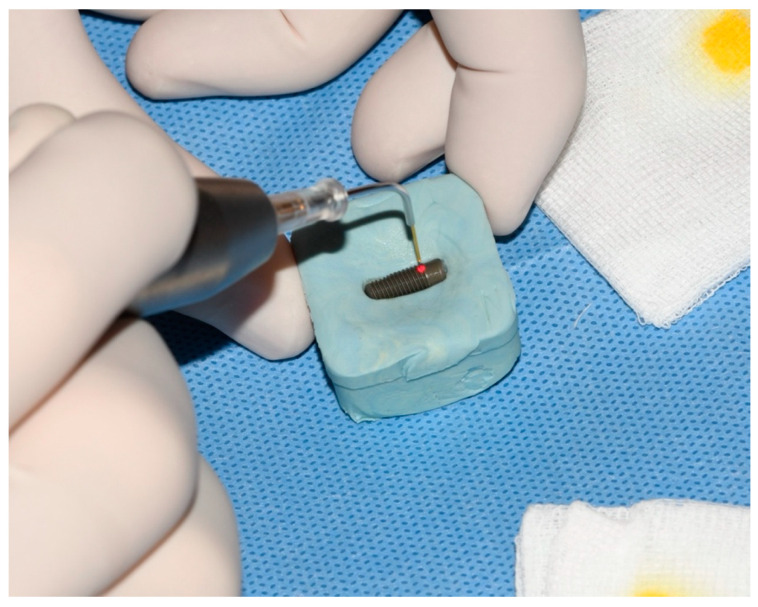
Treatment of implant surface with 445 nm diode laser for 60 s.

**Figure 6 bioengineering-09-00308-f006:**
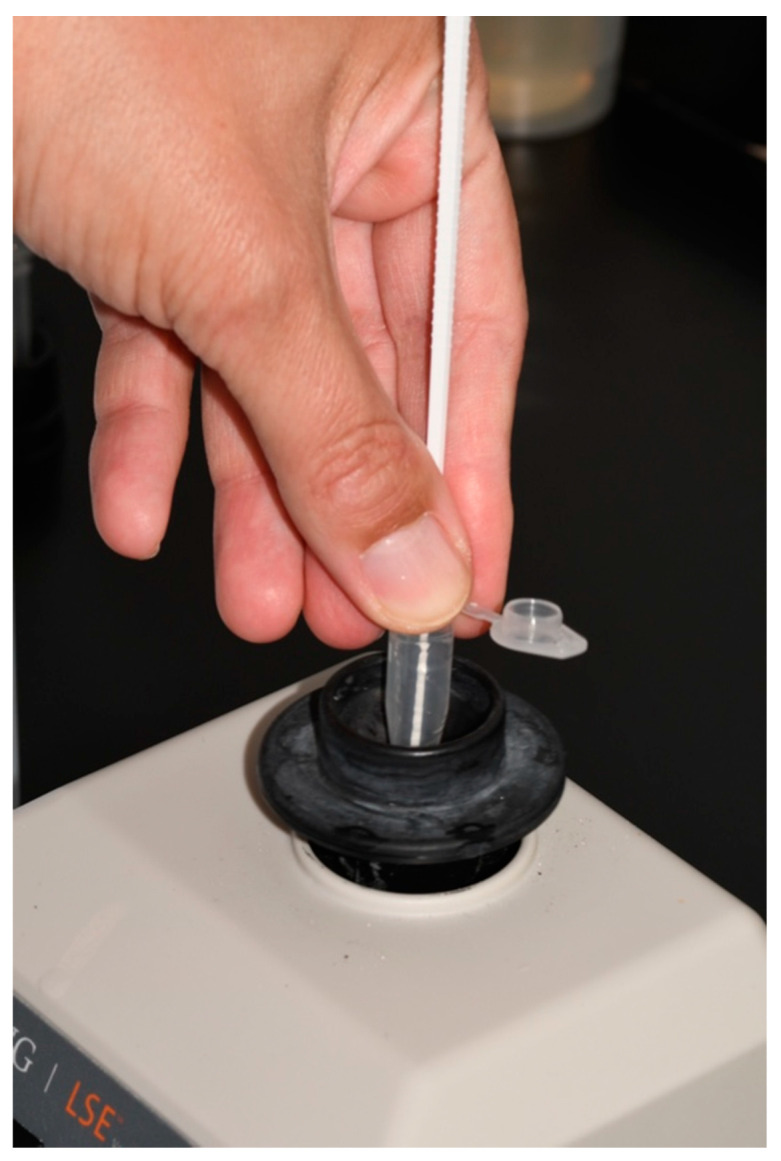
Vortexing of tube containing the infusion and plastic inoculating loop.

**Figure 7 bioengineering-09-00308-f007:**
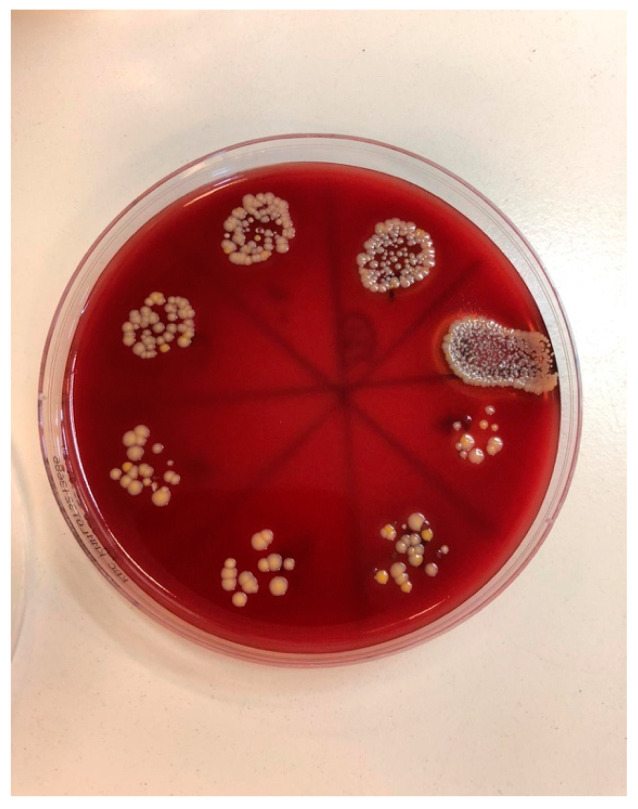
Pour plate technique, performed on blood agar after an incubation period of 48 h.

**Figure 8 bioengineering-09-00308-f008:**
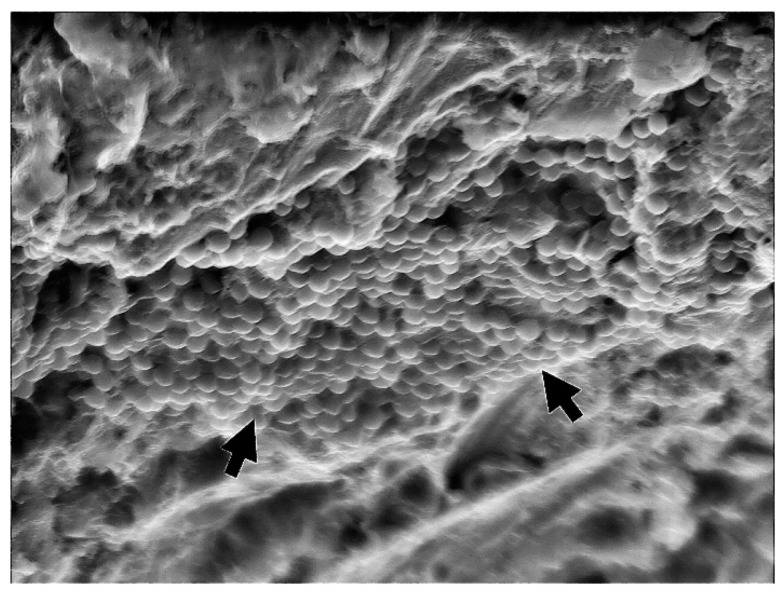
NC group—biofilm formation confirmed using SEM (5000×)—arrows showing microorganism biofilm formation.

**Figure 9 bioengineering-09-00308-f009:**
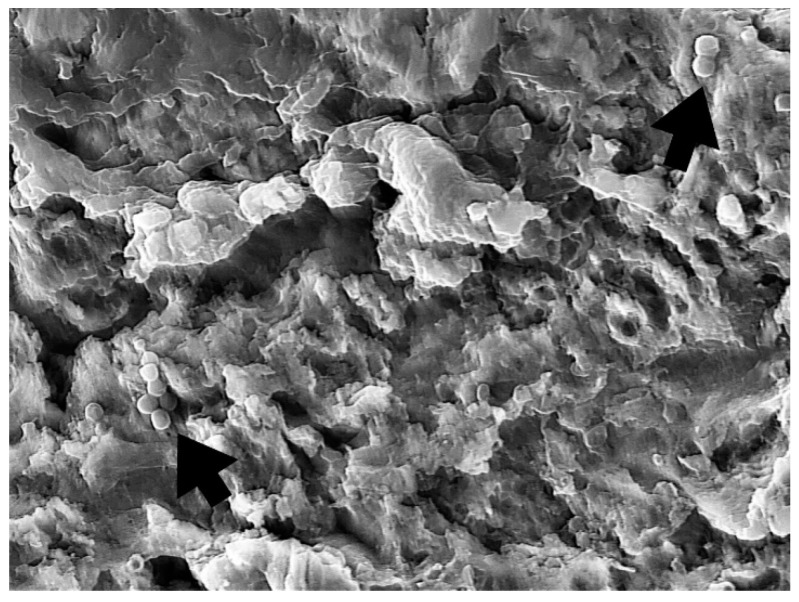
PC group—bacterial and fungi biofilm reduction after 0.2% CHX treatment (5000×). Microorganism cells can still be observed (arrow).

**Figure 10 bioengineering-09-00308-f010:**
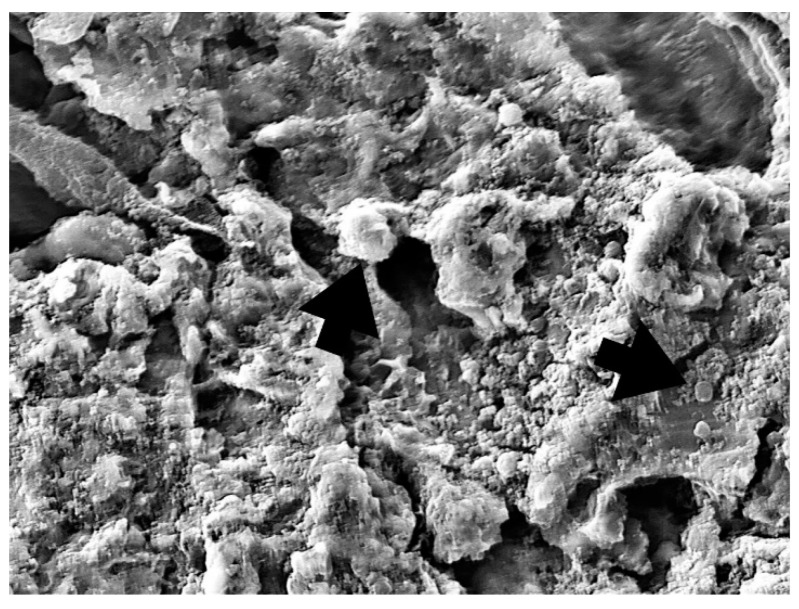
PDT1 group—bacterial and fungi reduction after 0.1% methylene blue and diode laser 660 nm treatment (5000×). Microorganism cells can still be observed (larger cell *C. Albicans*, smaller cell *S. Aureus*) (arrow).

**Figure 11 bioengineering-09-00308-f011:**
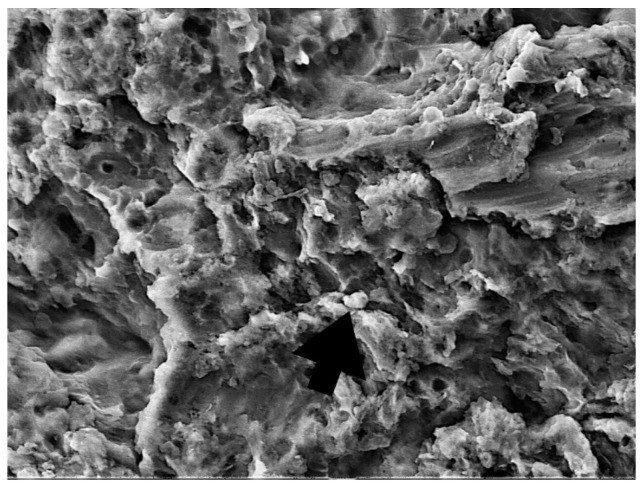
PDT2 group—bacterial and fungi reduction after 0.1% riboflavin and diode laser 445 nm treatment (3000×). Micro-organism cells can still be observed, but vitality of the cell cannot be discussed (arrow).

**Figure 12 bioengineering-09-00308-f012:**
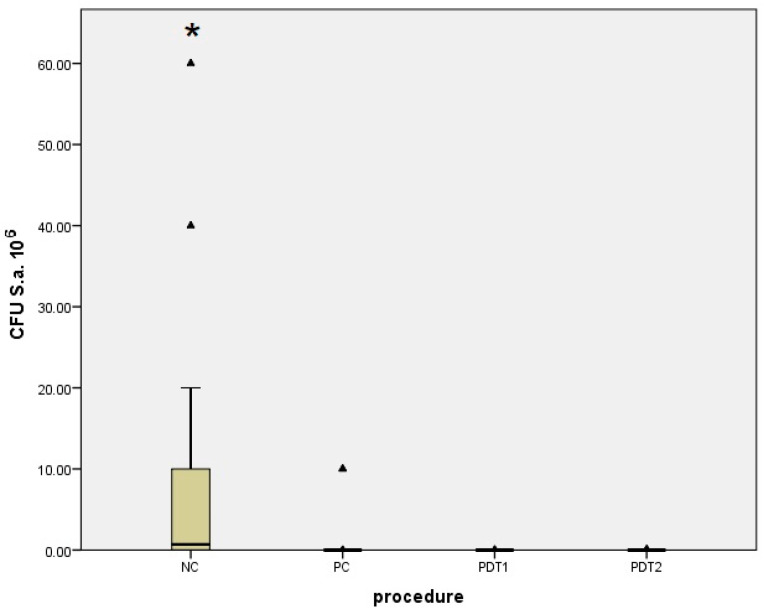
Difference in the number of *Staphylococcus aureus* CFUs between groups (NC, PC, PDT1, PDT2). ▴ Outliers; * Statistically significant difference.

**Figure 13 bioengineering-09-00308-f013:**
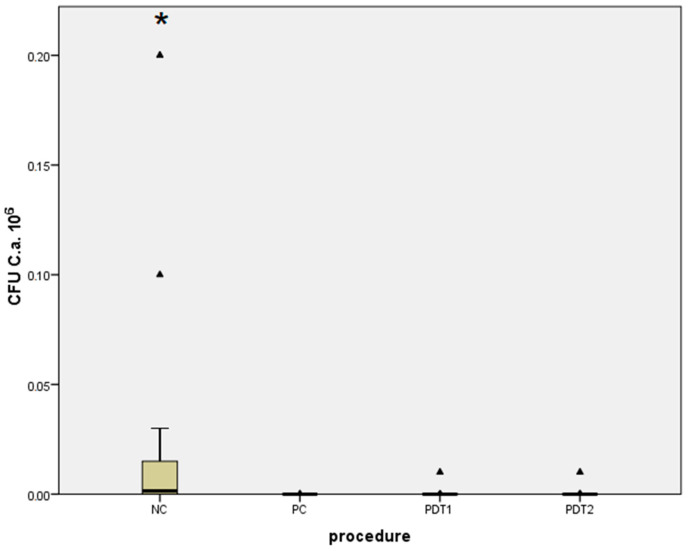
Difference in the number of *Candida albicans* CFUs between groups (NC, PC, PDT1, PDT2). ▴ Outliers; * Statistically significant difference.

**Table 1 bioengineering-09-00308-t001:** Difference in the number of *Staphylococcus aureus* CFUs between groups (NC, PC, PDT1, PDT2).

*Staphylococcus aureus*	Median (Interquartile Range)	Minimum–Maximum	Difference ^†^	95% CI	*p* *
PDT1	0 0 (0–5.5)	0–3 × 10^4^	3.187 × 10^6^	10^4^ to 10^7^	<0.001
NC	3.2 × 10^6^ (10^4^–1.5 × 10^7^)	10–10^8^			
PDT1	0 0 (0–0.51)	0–1	0	0 to 0	0.34
PC	0 (0–0)	0–10^7^			
PDT2	0.5 (0–1)	0–10^5^	3.15 × 10^6^	10^4^ to 10^7^	<0.001
NC	3.2 × 10^6^ (10^4^–1.5 × 10^7^)	10–10^8^			
PDT2	0.5 (0–1)	0–10^5^	0	−1 to 0	0.09
PC	0 (0–0)	0–10^7^			
NC	3.2 × 10^6^ (10^4^–1.5 × 10^7^)	10–10^8^	−4 × 10^5^	−1 × 10^7^ to −1 × 10^4^	<0.001
PC	0 (0–0)	0–10^7^			

CI, Confidence interval; *, Mann–Whitney U test; ^†^, Hodges–Lehmann median difference.

**Table 2 bioengineering-09-00308-t002:** Differences in *Candida albicans* CFUs between groups (NC, PC, PDT1, PDT2).

*Candida albicans*	Median (Interquartile Range)	Minimum–Maximum	Difference ^†^	95% CI	*p* *
PDT1	0 (0–1)	0–10^4^	10^3^	20 to 10^4^	<0.001
NC	1.5 × 10^3^ (20–1.5 × 10^4^)	0–2 × 10^5^			
PDT1	0 (0–1)	0–10^4^	0	0 to 0	0.15
PC	0 (0–0)	0–10^2^			
PDT2	0(0–0)	0–10^4^	10^3^	20 to 10^4^	0.001
NC	1.5 × 10^3^ (20–1.5 × 10^4^)	0–2 × 10^5^			
PDT2	0 (0–0)	0–10^4^	0	0 to 0	0.38
PC	0 (0–0)	0–10^2^			
NC	1.5 × 10^3^ (20–1.5 × 10^4^)	0–2 × 10^5^	−1450	−1 × 10^4^ to −1 × 10^2^	<0.001
PC	0 (0–0)	0–10^2^			

CI, Confidence interval; *, Mann–Whitney U test; ^†^, Hodges–Lehmann median difference.

**Table 3 bioengineering-09-00308-t003:** Difference in CFUs of PDT1 and PDT2.

Microorganism	Median (Interquartile Range)		Difference ^†^	95% CI	*p* *
	PDT1	PDT2			
*Staphylococcus aureus*	0 (0–5.5)	0.5 (0–1)	0	−1 to 0	0.55
*Candida albicans*	0 (0–1)	0 (0–0)	0	0 to 0	0.49

CI, Confidence interval; *, Mann–Whitney U test; ^†^, Hodges–Lehmann median difference.

## Data Availability

The data presented in this study are available on request from the corresponding author.

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
