# Peer review of "A Novel Technique for Disinfection Treatment of Contaminated Dental Implant Surface Using 0.1% Riboflavin and 445 nm Diode Laser—An In Vitro Study"

_bioengineering, 2022, doi:10.3390/bioengineering9070308_

Round 1
Reviewer 1 Report
The study is interesting. The authors assessed the efficiency of a new protocol of aPDT in disinfecting microbial biofilm on dental implants. The research is appropriately designed and the methods is adequately described. Though, a few concerns should be addressed before the manuscript can be published.
1. As many previous studies reported the efficiency of aPDT treatment consisting of a 660 nm diode laser with methylene blue or toluidine blue dye, why did the authors want to try a new protocol, the combination of a 445 nm diode laser and riboflavin solution? The purpose of using the new PDT treatment protocol should be explained in the part of Introduction more clearly. (Although the authors mentioned the disadvantage of methylene blue/toluidine blue such as discoloration in the esthetic zone in the part of Discussion)
2. Why did the authors choose to use SA and CA clinical isolates instead of ATCC strains like SA strain ATCC 25923? Also suggest adding patient information of clinical bacterial origin to the methodology, as well as isolation and identification procedures of SA and CA.
3. The picture of electron microscope is not clear enough, suggest replacing it with a higher quality picture.
4. What is the difference between a LED lamp and a laser lamp with the same wavelength? In the discussion, the authors mention " ~ and the light source being a LED curing lamp rather than a laser light ",as if a different light source would lead to different results, but in the next paragraph, the authors write "Although the light source was not a diode laser, as in our study, these results were in agreement with our study results from experiments conducted on dental implant surfaces contaminated with S. aureus." This seems inconsistent, please explain it more clearly.
Reviewer 2 Report
Dear Authors,
The article: A Novel Technique for Disinfection Treatment of Contaminated Dental Implant Surface Using 0.1% Riboflavin and 445 nm Diode Laser—An In Vitro Study
The study presented to me for evaluation is very interesting and well-planned.
As per the reviewer's obligation, I propose the following changes:
Introduction
1. “The bacterial composition in subgingival peri-implant pockets around the implant was found to be highly correlated with characteristics of the periodontal pocket microflora [4].” –
in this paragraph, expand the information, rather than give the differences in the composition of the subgingival flora between the peri-implant and periodontitis. it seems to be necessary because the study used Staphyloccocus aureus and Candida albicans strains, which are not important in the etiopathogenesis of periodontitis.
2. “Adjunctive therapies have been extensively investigated in contemporary dentistry, and many in vitro, animal, and clinical studies have already shown that noninvasive photodynamic therapy (PDT) can serve as a successful and safe adjunctive therapeutic protocol for the treatment of peri-implantitis [12–14]”.
I would add a sentence here regarding the precise information that PDT is adjunctive most effective in combination with prior mechanical cleaning of the implant thread surface.
Material and methods
1. “… or determination of significant differences in numerical values between measurements at the level of 0.05 and the power set to 0.8 (i.e., a large effect size of 0.8), the minimum required sample size was calculated as 20 subjects per group”.
great that the strength of the statistical test was calculated.
2. “The implants were randomly divided into four groups (n = 20), based on the planned surface treatment after contamination and biofilm formation, as follows:”
In the future, I would propose to add one more group in which the impact of rubbing with sterile cotton pellet without PS and CHX would be assessed.
3. “For the contaminated implants of the first photodynamic therapy group (PDT1), the surface was prepared using a similar procedure, a whereby 0.1% methylene blue solution (as a photosensitizer) was applied using a sterile syringe for 60 s followed by gentle washing with a sterile saline solution.”
Who is the producer of MB? Was it a commercial product intended for use in dentistry? And whether the incubation time of 60 s was well chosen - for bacteria OK, but for Candida, which is a eucaryote, the incubation time of 60 s may be a bit too short. More details in DOI 10.3390/ijms222010971. It’s about TBO but it’s the same group of dye.
4. “The surface was then dried with a sterile gauze and treated with laser (SiroLaser Blue, Dentsply Sirona, Bensheim, Germany) light at 660 nm (Q power = 100 mW), in continuouswave mode using an EasyTip 320 µm for 60 s moving in circles at approximately 1 mm from the implant surface. This process was carried out by a one skilled operator”
It is customary to specify in the laser settings parameters: spot size, fluence, energy density. especially that while exposing the implant thread, a scanning movement was performed. Then the energy set on the laser was transferred to some specific surface, therefore these values should be precisely given so that other scientists can compare your results with their own.
5. “The surface of the contaminated implants of the second photodynamic therapy group (PDT2) was treated with 0.1% riboflavin dye, applied using a sterile syringe and left for 60 s, followed by washing with sterile saline solution and gentle drying with a sterile gauze (Figure 4)”.
Who is the producer of ryboflavin? Was it a commercial product intended for use in dentistry? Was the 0.1% concentration optimal?
6. “After serial dilution (10X) suspension was inoculated to blood agar plate. After an incubation period of 48 h at 37 °C, the CFUs were counted (Figure 7)”.
This paragraph should be described in more detail. How many of these successive dilutions were there? What volume was plated on the blood agar plate? e.t.c
7. “Fig 8, 9, 10, 11” - it is a pity that the images at 5000x magnification as in figure 2 were not used here. It would be much more visible to the reader.
Results
1. Table 1, 2, 3 - the results could be presented much better in the form of a line and bar chart than in an unattractive table.
Discussion
Discussion It is interesting and comprehensive.
In the sentence “In the present study, 0.2% chlorhexidine (CHX) was used as a positive control, which was applied using cotton pellets using a brushing movement, thus simulating in vivo usage” - please remove the space before the comma.
Reviewer 3 Report
Please find the attached file

Round 2
Reviewer 1 Report
The revised manuscript is acceptable for publication
Reviewer 2 Report
Dear authors, the manuscript looks much better. Thank you for bringing in all the proposed changes.
Author Response
Dear reviewer,
thank you for your revisions.
Reviewer 3 Report
The light parameters need to be clearly demonstrated in Abstract.
Author Response
Please see the attachment

This manuscript is a resubmission of an earlier submission. The following is a list of the peer review reports and author responses from that submission.